# No Effect of Chronotype on Hunger or Snack Consumption during a Night Shift with Acute Sleep Deprivation

**DOI:** 10.3390/nu14071324

**Published:** 2022-03-22

**Authors:** Andrew M. Reiter, Gregory D. Roach, Charli Sargent

**Affiliations:** Appleton Institute for Behavioural Science, Central Queensland University, Goodwood, SA 5034, Australia; greg.roach@cqu.edu.au (G.D.R.); charli.sargent@cqu.edu.au (C.S.)

**Keywords:** snack, hunger, chronotype, circadian, misalignment, night, shift, sleep, deprivation

## Abstract

Night shift workers experience circadian misalignment and sleep disruption, which impact hunger and food consumption. The study aim was to assess the impact of chronotype on hunger and snack consumption during a night shift with acute sleep deprivation. Seventy-two (36f, 36m) healthy adults participated in a laboratory study. A sleep opportunity (03:00–12:00) was followed by a wake period (12:00–23:00) and a simulated night shift (23:00–07:00). Subjective measures of hunger, prospective consumption, desire to eat fruit, and desire to eat fast food were collected before (12:20, 21:50) and after (07:20) the night shift. Snack opportunities were provided before (15:10, 19:40) and during (23:50, 03:30) the night shift. A tertile split of the dim light melatonin onset (DLMO) distribution defined early (20:24 ± 0:42 h), intermediate (21:31 ± 0:12 h), and late chronotype (22:56 ± 0:54 h) categories. There were no main effects of chronotype on any subjective measure (*p* = 0.172–0.975), or on snack consumption (*p* = 0.420), and no interactions between chronotype and time of day on any subjective measure (*p* = 0.325–0.927) or on snack consumption (*p* = 0.511). Differences in circadian timing between chronotype categories were not associated with corresponding differences in hunger, prospective consumption, desire to eat fruit, desire to eat fast food, or snack consumption at any measurement timepoint.

## 1. Introduction

Approximately 20% of workers in industrialised economies work shifts outside of regular working hours [1,2]. Night shifts, which typically fall between 21:00 h and 08:00 h [3,4], are demanding because misalignment between the internal body clock and the imposed sleep–wake cycle impairs night-time alertness and day-time sleep [5,6,7,8,9,10]. Both circadian misalignment and sleep disruption impact hunger, quantity and quality of food consumption, and metabolic functioning [11,12]. Circadian misalignment is associated with increased consumption and reduced satiety [13], and sleep disruption is associated with increased appetite, hunger, consumption, and preference for foods that are higher in fat and sugar [12,14,15,16]. Circadian misalignment and sleep disruption are two of several factors proposed to contribute to the higher rates of obesity, cardiovascular disease, diabetes, and metabolic disorders displayed by shift workers compared with day workers [17,18,19].

During night shifts, workers tend to snack rather than consume complete meals [20]. Snacking behaviour appears to be influenced by both homeostatic sleep pressure and circadian processes. Energy intake from snacks is greater with restricted sleep (5.5 h/24 h) than with normal sleep (8.5 h/24 h) [21], suggesting elevated sleep pressure increases snack consumption. A simulated shift work study under forced desynchrony conditions found hunger, prospective consumption, and frequency of snack consumption were greater with severely restricted sleep (4 h/24 h) than with moderately restricted sleep (6 h/24 h) [12,22]. Furthermore, the same study reported a circadian effect, with hunger, prospective consumption, and frequency of snack consumption each exhibiting a minimum around the nadir of core body temperature [12,22]. This evidence is comparable to forced desynchrony studies that have found appetites for sweet food, salty food, fruit, and food overall exhibit pronounced circadian rhythms, with evening peaks that decline to morning troughs ~2–3 h after the nadir of core body temperature; see, for example, [23].

Chronotype is an individual difference that reflects circadian phase and sleep–wake behaviour. Night shift appears to be more challenging for early chronotypes than for late chronotypes due their relatively advanced biological and sleep–wake rhythms [24,25,26,27,28,29,30,31]. Typically, early chronotypes have healthier diets than late chronotypes, and consume more fruit and vegetables, and less sugary foods, drinks, and alcohol [32,33,34]. However, as hunger and snacking behaviour exhibit circadian rhythms, differences in circadian timing between early and late chronotypes may result in different snacking behaviours during a night shift with acute sleep deprivation.

Therefore, the aim of this study was to examine the impact of chronotype on snacking behaviour during a simulated night shift with acute sleep deprivation. Over the course of the night shift, snack consumption was expected to increase due to accumulated sleep pressure and reduce due to circadian influences. However, the combined impact of both factors was less clear. The main focus was to assess whether differences in circadian timing between early and late chronotypes were associated with differences in snack consumption as sleep pressure accumulated during the night shift.

## 2. Materials and Methods

### 2.1. Participants

A convenience sample was sourced from data collected during a laboratory study conducted at the Appleton Institute in Adelaide, South Australia. Participants were 72 young, healthy adults (36 females, 36 males) with a mean (±SD) age of 23.1 (±3.6) years and body mass index (BMI) of 21.5 (±1.9) kg/m^2^, recruited by advertisements posted at hostels, student accommodation, and university campuses, and on casual employment websites. Screening involved completion of a general health questionnaire and an in-person interview. Key inclusion criteria included age (18–30 years), measured BMI (18–25 kg/m^2^), and good physical and mental health. Key exclusion criteria included smoking, use of medications (excluding oral contraceptives), use of recreational drugs, excessive alcohol or caffeine consumption, excessive exercise, and shift work or transmeridian travel in the month prior to the study. The study was conducted according to the guidelines of the Declaration of Helsinki and approved by the Central Queensland University Human Research Ethics Committee (Approval Number: H16/06-168, October 2016). Participants provided written informed consent and were financially compensated with an honorarium of AUD 1080.

### 2.2. Design

A mixed design was used to assess the effects of time of day and chronotype on hunger and snack consumption before and during a single night shift. The within-subjects factor was time of day, and the between-subjects factor was chronotype (early, intermediate, late). Chronotype categorisations were objectively determined from dim light melatonin onset (DLMO), the gold standard biological marker of circadian phase [35]. DLMO-based chronotype categorisation allowed the hunger and snacking behaviour of early, intermediate, and late circadian phase groups with the same prior wakefulness (i.e., sleep pressure) to be compared. Four snack opportunities were provided: two during the day before the night shift, the third shortly after the start of the shift, and the fourth around the middle of the shift. The dependent variables were subjective hunger, prospective consumption, desire to eat fruit, desire to eat fast food, and snack consumption described in the Measures section.

### 2.3. Procedure

Data were collected during the first 3 days of a 10-day simulated shift work laboratory study (Figure 1). All participants experienced identical experimental conditions until one hour after the end of the first night shift (i.e., 08:00 on Day 3). In the week before the study, participants were instructed to maintain normal sleep habits, wear an activity monitor, and complete a sleep diary. During the study, participants were provided with standardised meals (i.e., breakfast, lunch, dinner). Participants were restricted from eating ad libitum, but water was available without restriction. Snack opportunities were provided between meals, and participants chose and consumed their snacks in their bedrooms to ensure their behaviour was not affected by others. Participants who selected a snack were given 10 min to consume the snack, and were not allowed to share their snack with other participants or keep their snack for later consumption.

Following a nine-hour sleep opportunity on Day 1, participants were provided with training and free time on Day 2. A baseline snack opportunity was provided at 16:00 (prior wake = 8.00 h), and baseline measures of hunger, prospective consumption, desire to eat fruit, and desire to eat fast food were collected before dinner at 19:00 (prior wake = 11.00 h). Saliva was collected hourly in dim light (<10 lux) from 19:00 until 03:00. The first saliva sample at 19:00 was collected just before dinner. Approximately twenty minutes before saliva was to be collected, participants gently rinsed their mouths with water, and were instructed to refrain from eating and drinking, and to remain seated. Saliva was collected from cotton swabs rolled in participants’ mouths for approximately 2–3 min, which were refrigerated, centrifuged, and frozen at −20 °C. Immediately after saliva sampling, participants were provided with a nine-hour sleep opportunity.

On Day 3, participants were kept awake (12:00–23:00) prior to an eight-hour simulated night shift (23:00–07:00). Subjective hunger, prospective consumption, desire to eat fruit, and desire to eat fast food were measured before pre-shift breakfast at 12:20 (prior wake = 0.33 h), before dinner at 21:50 (prior wake = 9:83 h), and before breakfast after the night shift at 07:20 (prior wake = 19.33 h). Snack opportunities were provided at 15:10 (prior wake = 3.17 h) and 19:40 (prior wake = 7.66 h), and during the night shift at 23:50 (prior wake = 11.83 h) and at 03:30 (prior wake = 15.5 h).

### 2.4. Measures

#### 2.4.1. Circadian Phase

Dim light melatonin onset (DLMO) was determined from saliva collected using cotton swabs (Salivette; Sarstedt, Nümbrecht, Germany). Melatonin concentration was measured by 4.3 pM direct radioimmunoassay, using reagents from Buhlmann Laboratories AG (Allschwill, Switzerland). DLMO was defined as the time melatonin concentration reached and stayed above a fixed threshold of 10 pM for at least two subsequent samples [36]. A higher relative threshold equal to the mean of the first three melatonin concentration values plus two standard deviations of those values was used for one participant whose melatonin concentration was above 10 pM for all samples [36]. The time of melatonin onset was estimated by linear interpolation between sample times immediately before and after concentration exceeded the threshold.

#### 2.4.2. Hunger, Prospective Consumption, Desire to Eat Fruit, and Desire to Eat Fast Food

Subjective hunger, prospective consumption, desire to eat fruit, and desire to eat fast food were measured using 100 mm visual analogue scales (VAS) that were marked in response to a question related to each construct. Visual analogue scales are reliable for within-subjects designs under controlled conditions, sensitive to experimental manipulation, and provide useful comparisons with food, energy, and nutrient measures [37,38]. Hunger was measured by the question: “How hungry do you feel right now?”, with the anchors “not at all” and “extremely”. Prospective consumption was measured by the question: “How much could you eat right now?”, with the anchors “almost nothing” and “a lot”. Desire to eat fruit was measured by the question: “How much would you enjoy eating fruit right now?”, with the anchors “not at all” and “very much”. Desire to eat fast food was measured by the question: “How much would you enjoy eating fast foods right now? (e.g., pizza, hamburgers, fried chicken)”, with the anchors “not at all” and “very much”. The score for each construct was in units equivalent to the number of millimetres measured to the right of the left anchor point (0–100), with higher scores indicating greater hunger, prospective consumption, desire to eat fruit, or desire to eat fast food.

#### 2.4.3. Snack Consumption

Participants could select one of four snack choices: (1) two servings of diced peaches or diced peaches and pears (total average energy = 586 kJ); (2) one serving of vanilla or mixed berry yoghurt (average energy = 595 kJ); (3) one apricot or mixed fruit muesli bar (average energy = 490 kJ); or (4) no snack (energy = 0 kJ). Snack choices were manually recorded, and the corresponding energy consumed was calculated.

#### 2.4.4. Statistical Analyses

IBM SPSS Statistics for Windows, Version 26.0 (Armonk, NY, USA) was used for all analyses. The Kolmogorov–Smirnov test was used to confirm normality of the DLMO distribution. Consistent with the approach recommended for comparing chronotypes based on mid-sleep times derived from MCTQ within a sample [39], participants with DLMO in the first, second, and third tertiles of the DLMO distribution were categorised as early, intermediate, and late chronotype, respectively. The effect of chronotype on baseline hunger, prospective consumption, desire to eat fruit, desire to eat fast food, and snack consumption were assessed by one-way between-subjects ANOVA. The effects of time of day and chronotype on hunger, prospective consumption, desire to eat fruit, desire to eat fast food, and snack consumption before, during, and immediately after the simulated night shift were assessed by two-way mixed factorial ANOVAs with one within-subjects factor (time of day) and one between-subjects factor (chronotype: early, intermediate, late). If Mauchly’s test indicated assumptions of sphericity were violated, degrees of freedom were corrected using Greenhouse–Geisser (ε < 0.75) or Huynh–Feld (ε > 0.75) estimates of sphericity. Statistical significance was determined using an alpha level of 0.05, with Bonferroni corrections applied to post-hoc comparisons of means.

## 3. Results

One participant withdrew from the study during saliva collection, and DLMO and measures of hunger, prospective consumption, desire to eat fruit, desire to eat fast food, and snack consumption were available for the remaining 71 participants. DLMO spanned a ~5.5 h range (19:12 to 00:47), and the Kolmogorov–Smirnov test confirmed the DLMO distribution was normal, *D*(71) = 0.10, *p* = 0.076. Participants with DLMO in the first tertile (earlier than or equal to 21:11) were categorised as early chronotype (*n* = 24), participants with DLMO in the third tertile (later than or equal to 21:54) were categorised as late chronotype (*n* = 24), and the remaining participants were categorised as intermediate chronotype (*n* = 23). Mean (*SD*) DLMO for the early, intermediate, and late chronotype categories were 20:24 (0:42), 21:31 (0:12), and 22:56 (0:54), respectively.

At baseline, there was no effect of chronotype on hunger (*p* = 0.42), prospective consumption (*p* = 0.35), desire to eat fruit (*p* = 0.38), desire to eat fast food (*p* = 0.65), or snack consumption (*p* = 0.60) (Figure 2). Before and immediately after the simulated night shift, there was no main effect of chronotype, and no interaction between chronotype and time of day, on hunger, prospective consumption, desire to eat fruit, desire to eat fast food, or snack consumption (Table 1, Figure 2). However, there were main effects of time of day on prospective consumption, desire to eat fruit, desire to eat fast foods, and snack consumption (Table 1, Figure 2). Prospective consumption was lower at post-shift breakfast than at both pre-shift breakfast (*p* = 0.017) and pre-shift dinner (*p* = 0.002). Desire to eat fruit was greater at pre-shift breakfast than at both pre-shift dinner (*p* < 0.001) and post-shift breakfast (*p* < 0.001). Desire to eat fast food was greater at pre-shift dinner than at both pre-shift breakfast (*p* < 0.001) and post-shift breakfast (*p* < 0.001). Snack consumption was similar at each of the two pre-shift snack opportunities (*p* = 0.061), was greater at the first night shift snack opportunity than at the second pre-shift snack opportunity (*p* = 0.015), and was similar at each of the two night shift snack opportunities (*p* = 0.666).

## 4. Discussion

The aim of this study was to assess the impact of chronotype on hunger and snack consumption during a simulated night shift with acute sleep deprivation. Chronotype did not impact any subjective measure or snack consumption at any timepoint before, during, or immediately after the night shift. These results suggest individuals with early, intermediate, and late circadian phase experience similar hunger, prospective consumption, desire to eat fruit, and desire to eat fast food, and exhibit similar snack consumption as homeostatic sleep pressure accumulates before and during a night shift.

The main focus of the present study was snack consumption during the night shift. As DLMO typically occurs ~7 h before the nadir of core body temperature [35], mean nadir of our early chronotype group (mean DLMO = 20:24) is estimated to have occurred at ~03:30 (near the mid-shift snack opportunity), and mean nadir of our late chronotype group (mean DLMO = 22:56) is estimated to have occurred at ~06:00 (near the end of the night shift). We found snack consumption was similar for all chronotype categories at the first night shift snack opportunity at 23:50 (prior wake = 11.83 h) and at the mid-shift snack opportunity at 03:30 (prior wake = 15.5 h). Furthermore, hunger, prospective consumption, desire to eat fruit, and desire to eat fast food were similar for all chronotype categories after the shift at 07:20 (prior wake = 19.33 h). Based on measures taken at these timepoints, early and late chronotypes exhibit similar hunger and snack consumption during a night shift as sleep pressure accumulates, despite differences in their circadian timing.

Forced desynchrony studies have shown hunger and frequency of snack consumption exhibit rhythms with evening peaks and early morning troughs. When sleep is unrestricted, the troughs occur ~3 h after the nadir of core body temperature [23]. However, when sleep is restricted, the troughs occur around the nadir of core body temperature [12,22]. The forced desynchrony protocol disentangles circadian and homeostatic influences by imposing sleep–wake cycles outside of the range of entrainment of the human circadian system, which systematically shift across the entire circadian cycle [40]. However, because the forced desynchrony protocol imposes regular sleep periods, it is not suitable for separating circadian and homeostatic influences associated with sleep deprivation. Therefore, in the present study, participants were influenced by both homeostatic sleep pressure and circadian phase which covaried with time of day, and the relative effects of these two factors on our measures could not be assessed.

A limitation of this study is that only sweet snacks were offered. As some shift workers believe sweet snacks will keep them awake [41], and sleep restriction is associated with greater likelihood of sweet snack consumption [12], future study designs could include savoury snack options for comparison with sweet snack options. Future protocols could also assess whether differences in hunger, prospective consumption, desire to eat fruit, desire to eat fast food, and snack consumption are evident with greater sleep deprivation, and include more frequent measures. For example, increasing the duration of wake prior to the start of the night shift from 11 h to 16 h would simulate a scenario of no daytime sleep, a situation which frequently occurs before a single night shift or the first of a series of consecutive night shifts [42]. It would also be beneficial to assess differences in snack consumption between chronotypes near the estimated late chronotype circadian nadir at ~06:00, where late chronotypes may exhibit their minimum snack consumption and early chronotypes may show an increase in snack consumption.

A boundary condition that may have influenced the results of the present study is that our sample consisted of young, healthy adults with normal sleep-wake patterns. Our findings for hunger, prospective consumption, desire to eat fruit, desire to eat fast food, and snack consumption may not generalise to extremely early or late chronotypes, or to clinical populations that exhibit atypical circadian rhythms.

In conclusion, we found differences in circadian timing between our early, intermediate, and late chronotype groups were not associated with corresponding differences in subjective measures of hunger, prospective consumption, desire to eat fruit, desire to eat fast food, or snack consumption at any of our measurement timepoints. Before and during a night shift, individuals with DLMO in the range ~19:00 to ~01:00 with prior wake of up to ~19 h experienced similar hunger and exhibited similar snack consumption at all measurement timepoints.

## Figures and Tables

**Figure 1 nutrients-14-01324-f001:**
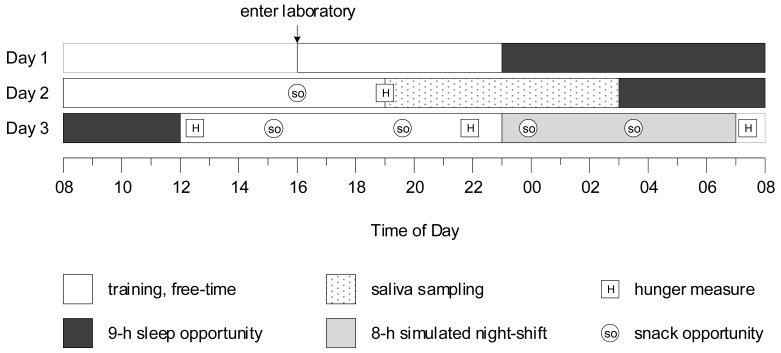
Protocol diagram, in which *y*-axis represents day of the protocol, and *x*-axis represents time of day. Baseline measures were collected on Day 2.

**Figure 2 nutrients-14-01324-f002:**
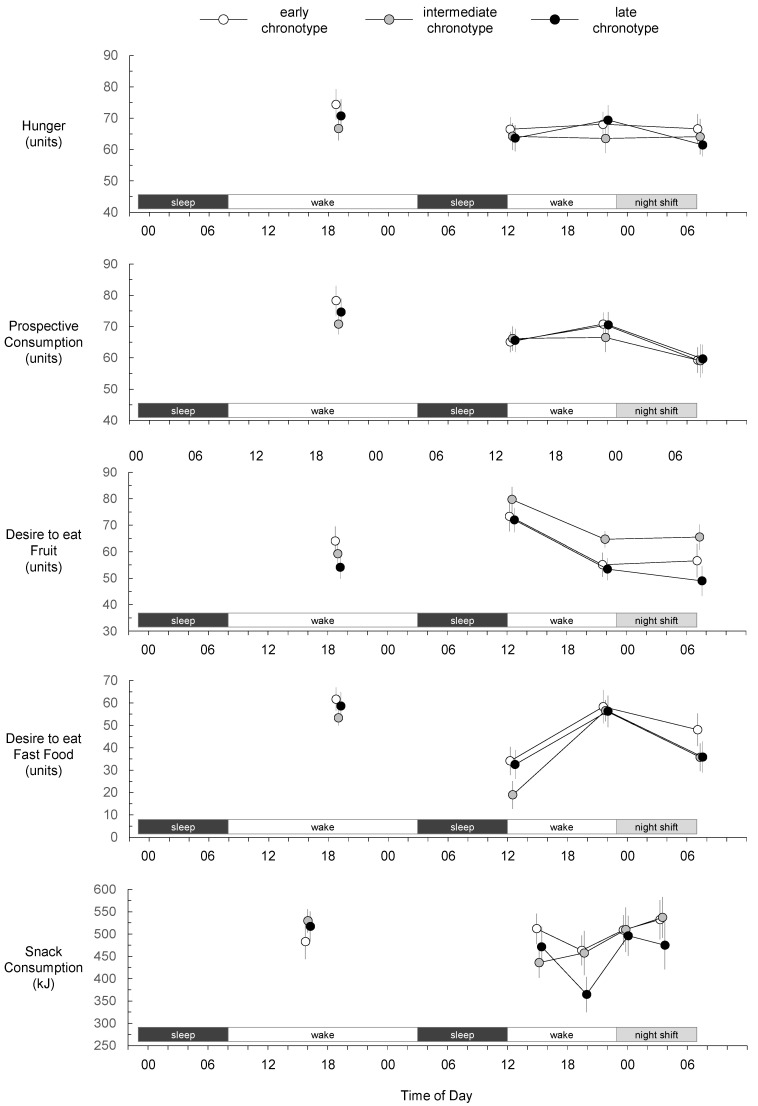
Hunger, prospective consumption, desire to eat fruit, desire to eat fast food, and snack consumption for early, intermediate, and late chronotypes. Baseline measures were collected during the first wake period. Mean values at each measurement have been offset to enhance interpretability. Error bars represent standard errors of the means.

**Table 1 nutrients-14-01324-t001:** Chronotype and time of day main and interaction effects on hunger, prospective consumption, desire to eat fruit, desire to eat fast food, and snack consumption before, during, and immediately after a simulated night shift.

	Chronotype Main Effect	Time of Day Main Effect	Chronotype × Time of Day Interaction Effect
Hunger	*F* (2.67) = 0.23, *p* = 0.799	*F* (1.8, 121.3) = 0.54, *p* = 0.564	*F* (3.6, 121.3) = 0.43, *p* = 0.767
Prospective Consumption	*F* (2.67) = 0.03, *p* = 0.975	*F* (2.134) = 6.77, *p* = 0.002	*F* (4.134) = 0.22, *p* = 0.927
Desire to eat Fruit	*F* (2.67) = 1.81, *p* = 0.172	*F* (2.134) = 26.05, *p* < 0.001	*F* (4.134) = 0.50, *p* = 0.734
Desire to eat Fast Food	*F* (2.67) = 0.74, *p* = 0.482	*F* (2.134) = 35.23, *p* < 0.001	*F* (4.134) = 1.17, *p* = 0.325
Snack Consumption	*F* (2.68) = 0.88, *p* = 0.420	*F* (2.72, 185.1)= 4.00, *p* = 0.011	*F* (5.4, 185.1) = 0.87, *p* = 0.511

Hunger, prospective consumption, desire to eat fruit, and desire to eat fast food were measured at three timepoints (12:20, 21:50, and 07:20.). Snack consumption was measured at four timepoints (15:10, 19:40, 23:50, and 03:30). *p* values for significant effects are shown in bold text.

## Data Availability

The data for this study are not currently publicly available, as they are part of a larger dataset that will be used for another purpose.

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
