# Peer review of "No Effect of Chronotype on Hunger or Snack Consumption during a Night Shift with Acute Sleep Deprivation"

_nutrients, 2022, doi:10.3390/nu14071324_

Round 1

Reviewer 1 Report

The study contributes to the growing field of chrono-nutrition, although the methodology is not without limitations, and this refers to limited snack options, regarding sort and number, and there is of course a potential discordance between desire and actual behavior.

Just one remark: Consider to state the amount of honorarium for participants, may be helpful for other researches

Author Response

We thank Reviewer 1 for their feedback. 

Reviewer 2 Report

This paper aim was to assess the impact of chronotype on hunger and snack consumption during a night shift with acute sleep deprivation. The topic is of interest since about 20% of employees work during the night and consequently are of increased risk for unhealthy eating habits. 

The paper is structured following the classical model for this type of papers – research article - including four parts: introduction; materials and methods; results and discussion, and conclusion. The  major components of the paper are balanced dimension-wise, presented logically and tightly linked to one another. The working methodology is tailored to the goals and, with very few exceptions, the materials and methods are presented appropriately.   The data yielded  following the laboratory determinations is presented in detail and in close correlation to some practical considerations. The overall paper quality is also complemented by an appropriate image gallery, consisting of 1 table and a 2 figure. The obtained results highlight both similarities to and differences from other studies.

Author Response

We thank Reviewer 2 for their feedback. 

No revisions to the manuscript are required.

Reviewer 3 Report

INTRODUCTION: Last paragraph, you described the aim of the study at the beginning, but after that you provide information of the study design that would be better in the corresponding section.  Also, in the final sentence you repeat the objective of the study

METHODS: The design is very interesting, but why only you collected data from the first 3 days of a 10-day study. What happened in the next 7 days?

Did you measure the sleep time in day 1 and 2? Could this affect the result?

RESULTS and DISCUSSION are ok

Author Response

We thank Reviewer 3 for their feedback. 

Round 2

Reviewer 3 Report

fine